# Beyond Information Gain: An Empirical Benchmark for Low-Switching-Cost Reinforcement Learning

**Shusheng Xu**[1], **Yancheng Liang**[1], **Yunfei Li**[1], **Simon S. Du**[2], **Yi Wu**[1,3]

[1] IIIS, Tsinghua University, [2] University of Washington, [3] Shanghai Qi Zhi Institute
{xuss20, liangyc19, liyf20}@mails.tsinghua.edu.cn
ssdu@cs.washington.edu, jxwuyi@gmail.com

*Reviewed on OpenReview:* `https://openreview.net/forum?id=Xq1sTZTQVm`

## Abstract

A ubiquitous requirement in many practical reinforcement learning (RL) applications is that the deployed policy that actually interacts with the environment cannot change frequently. Such an RL setting is called low-switching-cost RL, i.e., achieving the highest reward while reducing the number of policy switches during training. It has been a recent trend in theoretical RL research to develop provably efficient RL algorithms with low switching cost. The core idea in these theoretical works is to measure the *information gain* and switch the policy when the information gain is doubled. Despite of the theoretical advances, none of existing approaches have been validated empirically. We conduct the first empirical evaluation of different policy switching criteria on popular RL testbeds, including a medical treatment environment, the Atari games, and robotic control tasks. Surprisingly, although information-gain-based methods do recover the optimal rewards, they often lead to a substantially higher switching cost. By contrast, we find that a feature-based criterion, which has been largely ignored in the theoretical research, consistently produces the best performances over all the domains. We hope our benchmark could bring insights to the community and inspire future research. Our code and complete results can be found at *https://sites.google.com/view/low-switching-cost-rl*.

## 1 Introduction

Reinforcement Learning (RL) has been successfully applied to solve sequential-decision problems in many real-world scenarios, such as medical domains (Mahmud et al., 2018), robotics (Gu et al., 2017; Kalashnikov et al., 2021), hardware placements (Mirhoseini et al., 2017; 2020), and personalized recommendation (Zheng et al., 2018). When performing RL training in these scenarios, it is often desirable to restrict the agent from adjusting its policy too often due to the high costs and risks of deployment. For example, updating a treatment strategy in medical domains requires a thorough approval process by human experts (Almirall et al., 2012); in personalized recommendations, it is impractical to adjust the policy online based on instantaneous data and a more common practice is to aggregate data in a period before deploying a new policy (Zheng et al., 2018); in problems where we use RL to learn hardware placements (Mirhoseini et al., 2017), it is desirable to limit the frequency of changes to the policy since it is costly to launch a large-scale evaluation procedure on hardware devices like FPGA; for learning to control a real robot, it may be risky or inconvenient to switch the deployed policy when an operation is being performed. In these settings, it is a requirement that the deployed policy, i.e., the policy used to interact with the environment, could keep unchanged as much as possible. Formally, we would like our RL algorithm to both produce a policy with the highest reward and at the same time reduce the number of deployed policy switches (i.e., a low *switching cost*) throughout the training process.

In low-switching-cost RL, the central question is *how to design a criterion to decide when to update the deployed policy*. A good criterion should first ensure the *optimal* reward, then reduce the switching cost.

Low-switching-cost RL has recently become a hot topic in theoretical RL research. Many works have studied low-switching-cost RL and its simplified bandit setting extensively (Auer et al., 2002; Cesa-Bianchi et al., 2013; Bai et al., 2019; Ruan et al., 2020; Gao et al., 2021; Zhang et al., 2020a;b). The core notion in these theoretical works is *information gain*. Specifically, they update the deployed policy only if the measurement of information gain is doubled, which leads to optimality bounds for the final rewards. We suggest the readers refer to the original papers for detailed theoretical results. Algorithmic details will be presented in Section 3.2. However, no empirical evaluation has been conducted on whether these theoretically-guided criteria are in fact effective.

In this paper, we aim to provide systematic benchmark studies on different policy switching criteria empirically. We remark that our experiment scenarios are based on popular deep RL environments that are much more complex than the bandit or tabular cases studied in the existing theoretical works. Empirically, we find that, although information-gain-based criteria do achieve the optimal rewards, they perform poorly in reducing the switching cost. By contrast, a feature-based criterion, which is rarely studied by recent theoretical works, performs surprisingly well across all the experiment domains. We summarize our empirical findings and provide intuitive justifications for the feature-based method. We hope this benchmark could bring new insights to the community from both theoretical and practical perspectives and inspire future RL research towards this largely ignored feature-based direction.

Our contributions are summarized below.

- We conduct the first empirical study for low-switching-cost RL on environments that require modern RL algorithms, i.e., Rainbow (Hessel et al., 2018) and SAC (Haarnoja et al., 2018), including a medical environment, 56 Atari games[1] and 6 MuJoCo control tasks. We test theoretically guided switching criteria based on the information gain as well as other adaptive and non-adaptive criteria.

- We find that a feature-based criterion produces the best overall quantitative performance, which largely outperforms the theoretically guided ones based on information gain. This suggests a new research direction largely ignored by the existing literature.

- We summarize our empirical findings from a practical RL perspective and provide initial justifications and intuitions for the feature-based criterion.

## 2 Related Work

Low switching cost algorithms were first studied in the bandit setting (Auer et al., 2002; Cesa-Bianchi et al., 2013). Existing work on RL with low switching cost is mostly theoretical. To our knowledge, Bai et al. (2019) is the first work that studies this problem for the episodic finite-horizon tabular RL setting. Bai et al. (2019) gave a low-regret algorithm with an $O\left(H^3 S A \log\left(K\right)\right)$ local switching upper bound where $S$ is the number of stats, $A$ is the number of actions, $H$ is the planning horizon and $K$ is the number of episodes the agent plays. The upper bound was improved in (Zhang et al., 2020b;a). Recently, APEVE (Qiao et al., 2022) achieves a switching cost of $O\left(H S A \log\log\left(K\right)\right)$, which estimates the transition kernel, and adopts policy elimination by evaluating all policies using the estimated transition kernel. However, evaluating all policies is impossible in many real applications, and it is also non-trivial to estimate the transition kernel in many complex environments (model-based RL). In this paper, we focus on model-free RL and the design of the switching criterion, which is compatible with most existing DRL algorithms.

Offline RL (or Batch RL) can be viewed as a close but parallel variant of low-switching-cost RL, where the policy does not interact with the environment at all and therefore does not incur any switching cost. Offline RL methods typically learn from a given dataset (Lange et al., 2012; Levine et al., 2020) on a variety of domains including education (Mandel et al., 2014), dialogues (Jaques et al., 2019) and robotics (Kumar et al., 2020). However, a previously provided *high-quality* dataset, which is typically generated by experts or a well trained policy, may not always be feasible in many practical situations. In contrast with offline

---

[1]There are a total of 57 Atari games. However, only 56 of them (excluding the "surround" game) are supported by the atari-py package, which we adopt as our RL training interface.

RL, which optimizes reward subject to strong assumptions, i.e., a given dataset and zero policy switch, low-switching-cost RL aims to *reduce the switching cost while maintaining similar sample efficiency and final reward* compared to its unconstrained RL counterpart without any additional algorithmic assumptions. In theoretical literature, offline RL Antos et al. (2008); Tosatto et al. (2017); Xie & Jiang (2020); Wang et al. (2020) and low-switching-cost RL are also two parallel fields.

Some works interpolate offline and online RL, i.e., semi-batch RL (Singh et al., 1995; Lange et al., 2012), which update the policy many times on a large batch of transitions. However, reducing the switching cost during training is not their focus. Matsushima et al. (2021) is perhaps the most related work to us, which repeats offline training for a fixed (i.e., 10) iterations. In each deployment iteration, the proposed algorithm collects transitions using a fixed deployed policy, trains an ensemble of transition models and updates a new policy for the next iteration. However, even though the proposed model-based RL method in Matsushima et al. (2021) outperforms a collection of offline RL baselines, the final rewards are still substantially lower than online SAC even after consuming an order of magnitude more training samples. In our setting, an effective policy switching criterion should preserve comparable overall sample efficiency and the final rewards to online RL. There are also works that aim to reduce the interaction frequency with the environment rather than the switching cost (Gu et al., 2017; Hu et al., 2021), which are parallel to our focus.

## 3 Reinforcement Learning with Low Switching Cost

### 3.1 Notation

**Markov Decision Process:** We consider the Markov decision model $(\mathcal{S}, \mathcal{A}, \gamma, r, p_0, P)$, where $\mathcal{S}$ is the state space, $\mathcal{A}$ is the action space, $\gamma$ is the discounted factor, $r : \mathcal{S} \times \mathcal{A} \to \mathbb{R}$ is the reward function, $p_0$ is the initial state distribution, and $P(x'|x, a)$ denotes the transition probability from state $x$ to state $x'$ after taking action $a$. A policy $\pi : \mathcal{S} \to \mathcal{A}$ is a mapping from a state to an action. An episode starts with an initial state $x_0 \sim p_0$. At each step $h$ in this episode, the agent chooses an action $a_h$ from $\pi(x_h)$ based on the current state $x_h$, receives a reward $r(x_h, a_h)$ and moves to the next state $x_{h+1} \sim P(\cdot|x_h, a_h)$. We assume an episode will always terminate, so each episode $e = \{(x_h^e, a_h^e)|0 \le h \le H_e\}$ will always have a finite horizon $H_e$. The goal of the agent is to find a policy $\pi^*$ which maximizes the discounted expected reward, $\pi^\star = \arg\max_\pi \mathbb{E}_e \left[ \sum_{h=0}^{H_e} \gamma^h r(x_h^e, a_h^e) \right]$. Ideally, we also want the agent to consume as few training samples as possible to learn $\pi^\star$.

**Low-switching-cost RL:** In standard RL, a transition $(x_h, a_h, x_h)$ is always collected by a single policy $\pi$. Therefore, whenever the policy is updated, a switching cost is incurred. In low-switching-cost RL, we have two separate policies, a deployed policy $\pi_{\text{dep}}$ that interacts with the environment, and an online policy $\pi_{\text{onl}}$ that is trained by the underlying RL algorithm. These policies are parameterized by $\theta_{\text{dep}}$ and $\theta_{\text{onl}}$ respectively. Suppose that we totally collect $K$ samples during the training process, then at each transition step $k$, the agent is interacting with the environment using a deployed policy $\pi_{\text{dep}}^k$. After the transition is collected, the agent can decide whether to update the deployed $\pi_{\text{dep}}^{k+1}$ by the online policy $\pi_{\text{onl}}^{k+1}$, i.e., replacing $\theta_{\text{dep}}$ with $\theta_{\text{onl}}$, according to some switching criterion $\mathcal{J}$. Accordingly, the switching cost is defined by the number of different deployed policies throughout the training process, namely:

$$C_{\text{switch}} := \sum_{k=1}^{K-1} \mathbb{I}\{\pi_{\text{dep}}^{k-1} \ne \pi_{\text{dep}}^k\}. \tag{1}$$

In standard RL, $C_{\text{switch}}$ equals the number of policy updates during the training process, which can be millions when using DRL algorithms. Such a large $C_{\text{switch}}$ is unacceptable in the applications of medical domains and hardware placements. The goal of low-switching-cost RL is to design an effective algorithm that learns $\pi^*$ using $K$ samples while produces the smallest switching cost $C_{\text{switch}}$. Particularly in this paper, we focus on the design of the switching criterion $\mathcal{J}$, which is the most critical component that balances the final reward and the switching cost. The overall workflow of low-switching-cost RL is shown in Algorithm 1.

In the following content, we present a collection of policy switching criteria and techniques, including those inspired by the information gain principle (Sec. 3.2) as well as other non-adaptive and adaptive criteria (Sec. 3.3). All the discussed criteria are summarized in Algorithm 2 of Appendix B.

---

**Algorithm 1** General Workflow of Low-Switching-Cost RL

---

1: Initialize parameters $\theta_{\text{onl}}, \theta_{\text{dep}}$, an empty replay buffer $D$, $C_{\text{switch}} \leftarrow 0$
2: **for** k $\leftarrow 0$ to $K - 1$ **do**
3:      Select $a_k$ by $\pi_{\text{dep}}(x_k)$, execute action $a_k$ and observe reward $r_k$, state $x_{k+1}$
4:      Store $(x_k, a_k, r_k, x_{k+1})$ in $D$
5:      Update $\theta_{\text{onl}}$ using $D$ and an off-policy RL algorithm
6:      **if** $\mathcal{J}(\star) ==$ true **then**
7:          Update $\theta_{\text{dep}} \leftarrow \theta_{\text{onl}}$, $C_{\text{switch}} \leftarrow C_{\text{switch}} + 1$

---

### 3.2 Switching via Information Gain

Existing theoretical studies propose to switch the policy whenever the agent has gathered sufficient new information. Intuitively, if there is not much new information, then it is not necessary to switch the policy. The information gain is measured by the visitation count of each state-action pair or the determinant of the covariance matrix. We implement these two criteria as follows.

**Visitation-based Switching:** Following Bai et al. (2019), we switch the policy when visitation count of any state-action pair reaches an exponent (specifically $2^i, i \in \mathbb{N}$ in our experiments). Such exponential scheme is theoretically justified with bounded switching cost in tabular cases. However, it is not feasible to maintain precise visitations for high-dimensional states, following count-based exploration (Tang et al., 2017), we adopt a random projection function to map the states to discrete vectors by $\phi(x) = \text{sign}(A \cdot g(x))$, and then count the visitation to the hashed states as an approximation. $A$ is a fixed matrix with i.i.d entries from a unit Gaussian distribution $\mathcal{N}(0, 1)$ and $g$ is a flatten function which converts $x$ to a 1-dimensional vector.

**Information-matrix-based Switching:** Another algorithmic choice for achieving infrequent policy switches is based on the property of the feature covariance matrix (Ruan et al., 2020; Gao et al., 2021), i.e., $\Lambda_h^e = \sum_{e:H_e \geq h} \psi(x_h^e, a_h^e)\psi(x_h^e, a_h^e)^T + \lambda I$, where $e$ denotes a training episode, $h$ means the $h$-th timestep within this episode, and $\psi$ denotes a mapping from the state-action space to a feature space. For each episode timestep $h$, Abbasi-Yadkori et al. (2011) switch the policy when the determinant of $\Lambda_h^e$ doubles. However, we empirically observe that the approximate determinant computation can be particularly inaccurate for complex RL problems. Instead, we adopt an effective alternative, i.e., switch the policy when the least absolute eigenvalue doubles. In practice, we again adopt a random projection function to map the state to low-dimensional vectors, $\phi(x) = \text{sign}(A \cdot g(x))$, and concatenate them with actions to get $\psi(x, a) = [\phi(x), a]$.

### 3.3 Other Switching Criteria

The information-gain-based criteria are proposed under bandit or tabular cases theoretically, and our experiment scenarios are much more complex. We further investigate some other criteria empirically.

**Non-adaptive Switching Criterion:** This simplest strategy switches the deployed policy every $n$ timesteps, which we denote as "FIX_n" in our experiments. Empirically, we notice that "FIX_1000" is a surprisingly effective criteria which remains effective in most of the scenarios without hyperparameter tuning. So we primarily focus on "FIX_1000" as our non-adaptive baseline. In addition, We specifically use "None" to indicate the experiments without the low-switching-cost constraint where the deployed policy keeps synced with the online policy all the time.

**Policy-based Switching Criterion:** Another straightforward criterion is to switch the deployed policy when the action distribution produced by the online policy significantly deviates from the deployed policy. Specifically, in discrete action domains, we sample a batch of training states and count the number of states

where actions taken by the two policies differ. We switch the policy when the ratio of mismatched actions exceeds a threshold $\sigma_p$. For continuous domains, we use KL-divergence to measure the policy differences.

**Feature-based Switching Criterion:** Beyond directly consider the difference of action distributions, another possible solution for measuring the divergence between two policies is through the feature representation extracted by the neural networks. Hence, we consider a feature-based switching criterion that decides to switch policies according to the feature similarity between different Q-networks. Similar to the policy-based criterion, we first sample a batch of states $\mathbb{B}$ from the experience replay buffer, and then extract the features of all states with both the deployed Q-network and the online Q-network. It is noteworthy that there is an MLP with a single hidden layer between the feature representation we adopted and the final output of the Q-network. For a state $x$, the extracted features are denoted as $f_{\text{dep}}(x)$ and $f_{\text{onl}}(x)$, respectively. The similarity score between $f_{\text{dep}}$ and $f_{\text{onl}}$ on state $x$ is defined as

$$sim(x) = \frac{\langle f_{\text{dep}}(x), f_{\text{onl}}(x) \rangle}{||f_{\text{dep}}(x)|| \times ||f_{\text{onl}}(x)||}. \tag{2}$$

We then compute the averaged similarity score on the batch of states $\mathbb{B}$

$$sim(\mathbb{B}) = \frac{\sum_{x \in \mathbb{B}} sim(x)}{||\mathbb{B}||}. \tag{3}$$

With a hyper-parameter $\sigma_f \in [0, 1]$, the feature-based policy switching criterion changes the deployed policy whenever $sim(\mathbb{B}) \leq \sigma_f$.

### 3.4 Implementation

In this section, we summarize the implementation details of above criteria.

**Reset-Checking:** Empirically, we also find an effective implementation enhancement, which produces lower switch costs and is more robust across different environments: we *only* check the feature similarity when an episode *resets* (i.e., a new episode starts).

**Estimation of Feature Similarity & Action Difference:** For policy-based and feature-based criteria, we uniformly sample 512 from recent 10,000 transitions, and compare the action differences or feature similarities between the deployed policy and the online policy on these sampled transitions. We also tried other sample size and sampling method, and there is no significant difference.

**Switching Threshold:** For the switching threshold (i.e., the mismatch ratio $\sigma_p$ in policy-based criterion and parameter $\sigma_f$ in feature-based criterion), we perform a rough grid search and choose the highest possible threshold that still produces a comparable final policy reward. [2]

## 4 Experiments

In this section, we conduct experiments to evaluate different policy switching criteria on Rainbow DQN (Hessel et al., 2018) and SAC (Haarnoja et al., 2018). For discrete action spaces, we study the Atari games and the GYMIC testbed for simulating sepsis treatment for ICU patients which requires low switching cost. For continuous control, we conduct experiments on the MuJoCo (Todorov et al., 2012) locomotion tasks. The details of GYMIC, Atari and MuJoCo environments are introduced in Appendix C.

### 4.1 Evaluation Metric

For GYMIC and Atari games whose action spaces are discrete, we adopt Rainbow DQN to train the policy; for MuJoCo tasks with continuous action spaces, we employ SAC since it is more suitable for continuous action spaces. We evaluate the efficiency among different switching criteria in these environments. For the Atari and Mujoco environments, we conduct experiments over 5 seeds. For the GYMIC environment, the

---

[2]We list the hyper-parameter search space in Appendix C.

experiments are repeated over 10 seeds due to the high variance in this environment. Implementation details and hyper-parameters are listed in Appendix C.

We benchmark different policy switching criteria by measuring the accumulated reward as well as the switching cost in all the environments. We report the average results of all the Atari games and detailed results of 8 representative games in the main paper due to space limit. Results for every single Atari game can be found at our project website[3].

To better quantitatively measure the effectiveness of a policy switching criterion, we propose a new evaluation metric, **Reward-threshold Switching Improvement (RSI)**, which takes both the policy performance and the switching cost improvement into account. Specifically, suppose the standard online RL algorithm (i.e., the "None" setting) can achieve an average reward of $\hat{R}$ with switching cost $\hat{C}$[4]. Now, a low-switching-cost RL criterion $\mathcal{J}$ leads to a reward of $R_{\mathcal{J}}$ and reduced switching cost of $C_{\mathcal{J}}$ using the same amount of training samples. Then, we define RSI of criterion $\mathcal{J}$ as

$$RSI(\mathcal{J}) = \mathbb{I}\left[R_{\mathcal{J}} > \left(1 - \text{sign}(\hat{R})\sigma_{\text{RSI}}\right)\hat{R}\right] \log\left(\max\left(\frac{\hat{C}}{C_{\mathcal{J}}}, 1\right)\right), \tag{4}$$

where $\mathbb{I}[\cdot]$ is the indicator function and $\sigma_{\text{RSI}}$ is a reward-tolerance threshold indicating the maximum allowed performance drop with the low-switching-cost constraint applied. In our experiments, we choose a fixed threshold parameter $\sigma_{\text{RSI}} = 0.2$. Intuitively, when the performance drop is moderate (i.e., within the threshold $\sigma_{\text{RSI}}$), RSI computes the logarithm [5] of the relative switching cost improvements; while when the performance decreases significantly, the RSI score will be simply 0.

## 4.2 Results

We compare the performances of all the criteria presented in Sec. 3, including unconstrained RL ("None"), non-adaptive switching ("Fix_1000"), policy-based switching ("Policy"), feature-based switching ("Feature") and two information-gain variants, namely visitation-based ("Visitation") and information-matrix-based ("Info") criteria.

**GYMIC:** This medical environment is relatively simple, and all the criteria achieve similar learning curves as unconstrained RL as shown in Fig. 1. However, the switching cost of visitation-based criterion is significantly higher – it almost overlaps with "None". While the other information-gain variant, i.e., information-matrix-based criterion, performs much better in this scenario. Overall, feature-based criterion produces the most satisfactory switching cost without hurt to sample efficiency. GYMIC simulates the process of treating ICU patients. Each episode starts from a random state of a random patient, which only provides a binary reward (+15 or -15) at the end of each episode for whether the patient is cured or not. We notice that even a random policy can achieve an average reward of 13, which is very close to the maximum possible reward 15. Therefore, possible improvement by RL policies can be small in this case. However, we want to emphasize that despite similar reward performances, different switching criteria lead to significantly different switching costs.

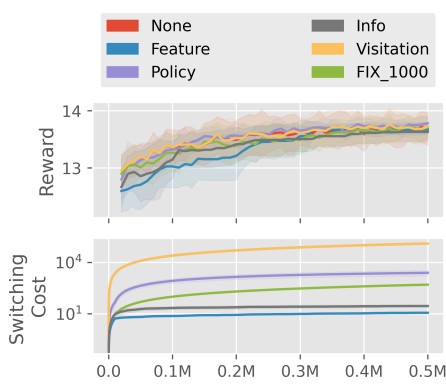

Figure 1: Results on GYMIC. *Top*: rewards vs. steps. *Bottom*: switching costs. The switching cost of "Visitation" almost overlaps with "None".

**Atari Games:**

We then compare the performances of different switching criteria in the more complex Atari games. The state spaces in Atari games are images, which are more complicated than the low-dimensional states in GYMIC. Fig. 2 shows the average reward and switching of different switching criteria across all the 56 games, where the feature-based solution leads to the best empirical performance. We also remark that the non-adaptive

---

[3]https://sites.google.com/view/low-switching-cost-rl

[4]We use $\hat{C}$ here since some RL algorithm may not update the policy every timestep.

[5]We also tried a variant of RSI that remove the log function. The results are shown in the appendix D.

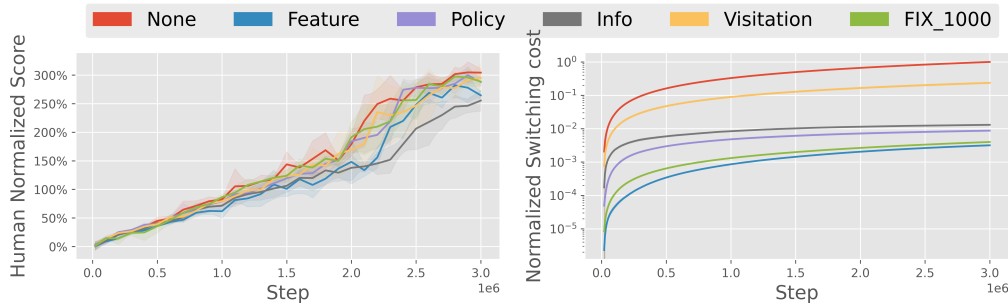

Figure 2: The average results on Atari games. We compare different switching criteria across 56 Atari games. *Left:human normalized reward. Right:* the average switching cost, which is normalized by the switching cost of "None" and shown in a log scale.

baseline is particularly competitive in Atari games and outperforms all other adaptive solutions except the feature-based one. We can observe that information-gain variants produce substantially more policy switches. while the feature-based solution switches less frequently.

In addition, we notice that the policy-based solution is particularly sensitive to $\sigma_p$ in order to produce desirable policy reward with low switching cost, while feature-based solution is easier to tune, which suggests that the neural network features may change more smoothly than the action distribution.

To validate this hypothesis, we visualize the action difference and feature difference of the unconstrained Rainbow DQN on the Atari game "Pong" throughout the training process in Fig. 3. Note that in this case, the deployed policy is synced with the online policy in every training step, so the difference is merely due to a single training update. However, even in this unconstrained setting, the difference of action distribution fluctuates significantly. By contrast, the feature change is much more stable.

**MuJoCo Control:** We evaluate the effectiveness of different switching criteria with SAC on all the 6 MuJoCo continuous control tasks. The results are shown in Fig. 4. In general, we can still observe that the feature-based solution achieves the lowest switching cost among all the baseline methods while the policy-based solution produces the most unstable training. Interestingly, although the non-adaptive baseline has a relatively high switching cost than the feature-based one, the reward curve has less fluctuation, which also suggests a future research direction on incorporating training stability into the switching criterion design.

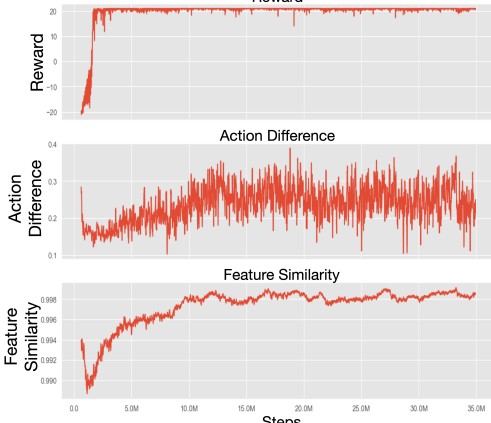

Figure 3: Action difference and feature similarity tested on Pong. Higher feature similarity or lower action difference implies that two policies are closer.

**Average RSI Scores:** We also report the RSI scores of different policy switching criteria on different domains. For each domain, we compute the average value of RSI scores over each individual task in this domain. The results are reported in Table 1, we can observe that the feature-based method consistently produces the best quantitative performance across all the 3 domains. [6]

| Avg. RSI | Feature | Policy | Info | Visitation | FIX_1000 |
|----------|---------|--------|------|------------|----------|
| GYMIC | **9.30** | 3.93 | 8.39 | 0.0 | 6.91 |
| Atari | **3.60** | 3.22 | 2.33 | 1.83 | 3.25 |
| Mujoco | **8.26** | 5.26 | 4.69 | 1.90 | 6.68 |

Table 1: RSI (Eq. 4, $\sigma_{\text{RSI}} = 0.2$) of different criteria over different domains. We take unconstrained RL (i.e., "None") performance as the RSI reference, so the RSI value for "None" is always zero.

---

[6]We list the results of $\sigma_{\text{RSI}} = 0.2$ in Table 1, and also evaluate RSI using other $\sigma_{\text{RSI}}$. See Appendix D for details.

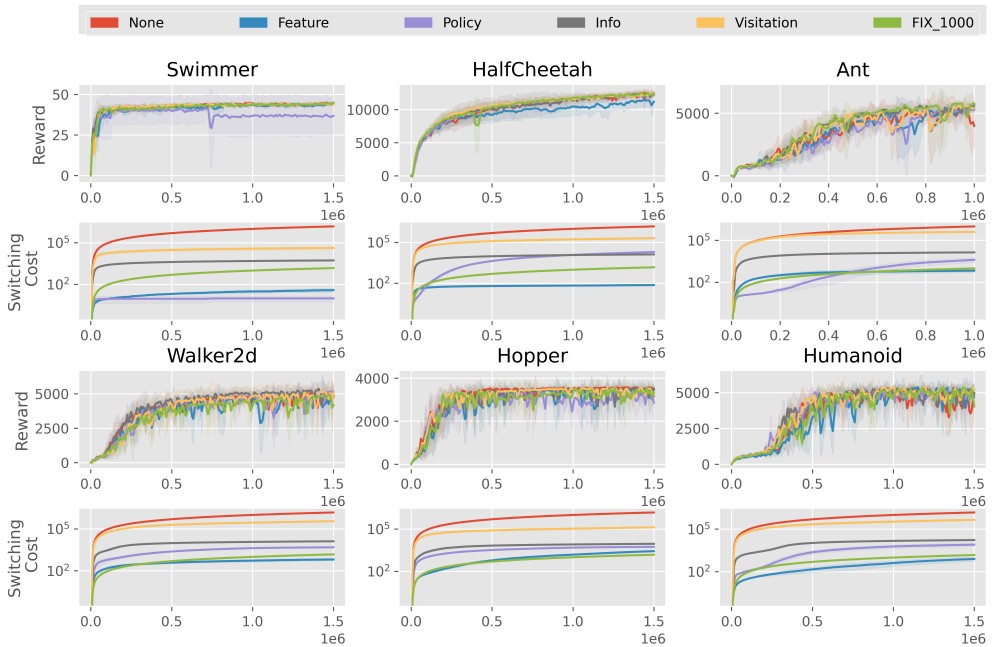

Figure 4: The results on MuJoCo tasks.

In addition, we apply a *t*-test for the switching costs among different criteria. The testing procedure is carried out for each pair of methods over all the experiment trials across all the environments and seeds. For Atari and MuJoCo, the results show that there are significant differences in switching cost between any two criteria (with *p*-value $< 0.05$). For GYMIC, "None" and "Visited" is the only pair with no significant difference. It is also worth mentioning that RSI for "Visitation" for GYMIC is 0, which also shows the switching costs of "Visitation" and "None" are nearly the same.

**Comparison with Offline RL:** We also compare the low-switching-cost setting with offline RL empirically. In Fig. 5, We adopt the implementations of a popular offline RL algorithm CQL (Kumar et al., 2020) and its online counterpart QR-DQN (Agarwal et al., 2020b) provided by the authors.[7] For CQL, we adopt the same dataset with the author.[8] We apply the feature-based switching criteria to QR-DQN and conduct the experiments on 4 Atari games considered in CQL. From Fig. 5, we observe that feature-based criterion can also reduce the switching cost while maintaining the performance. CQL learns from a fixed dataset, it switches just once and doesn't aim to recover the online performance, parallel to our focus.

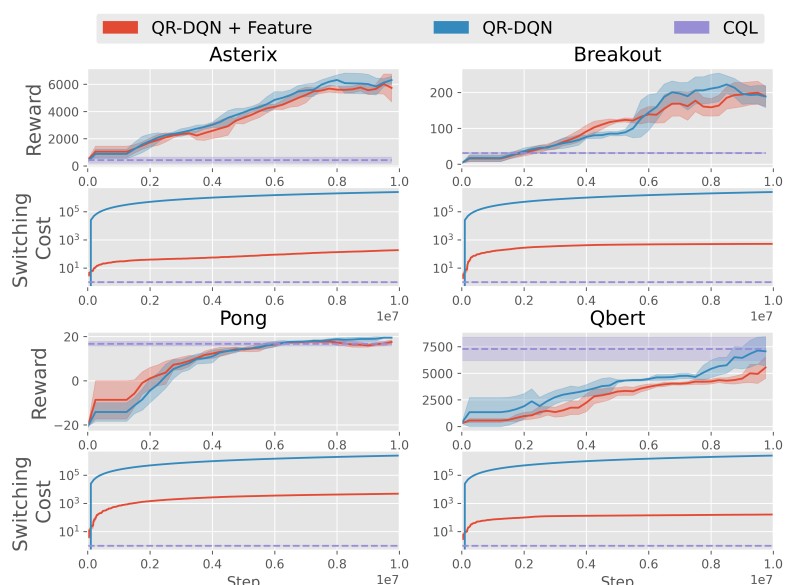

Figure 5: Comparison with offline RL. Offline RL does not aim to recover the online performance.

---

[7]https://github.com/aviralkumar2907/CQL

[8]The dataset of Atari comes from https://github.com/google-research/batch_rl

In Fig. 6, we further compare the low-switching-cost setting against several recent offline RL methods (Fujimoto & Gu, 2021; Fujimoto et al., 2019; Kumar et al., 2020) on MuJoCo tasks. We use the author-provided implementations for the offline algorithms[9][10] and evaluate these methods on the D4RL (Fu et al., 2020) benchmark. For TD3 (online) and TD3+Feature, we remove the behavior cloning regularization term and the feature-normalizing process in the codebase of TD3+BC, which are designed for offline training. The results in Fig. 6 indicate that the performances of offline methods are *highly* dependent on the dataset's quality. No offline algorithm obtains comparable performances with online training on the "random" or "medium-replay" dataset. However, the low-switching-cost setting does not need a dataset. It can recover online training performances and reduce switching costs by orders of magnitude.

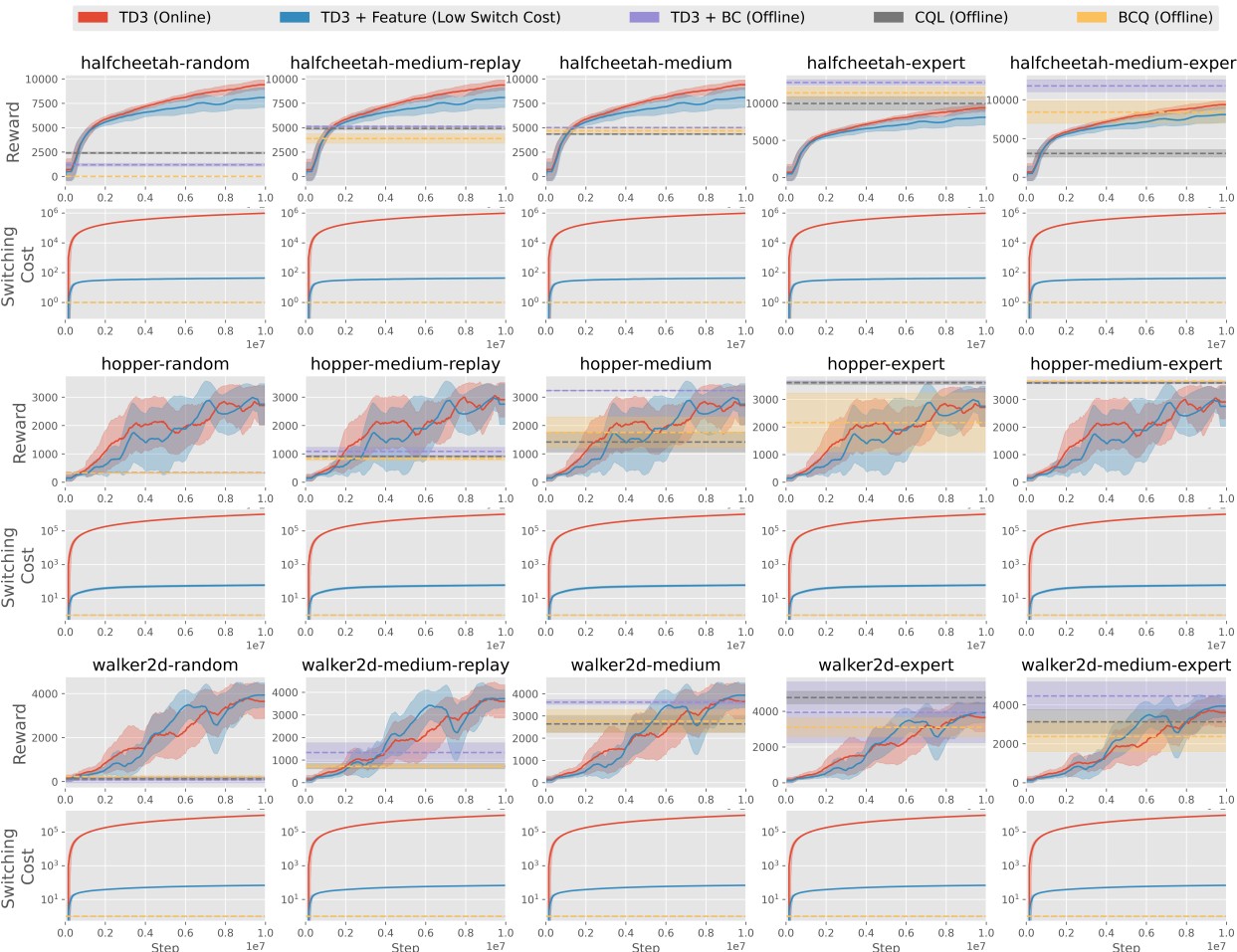

Figure 6: Comparison with offline RL on the D4RL benchmark. The performances of offline methods are *highly* dependent on the quality of the dataset. "Random" uses 1M samples from a randomly initialized policy. "Expert" uses 1M samples from a policy trained to completion with SAC. "Medium" uses 1M samples from a policy trained to approximately 1/3 the performance of the expert. "Medium-Replay" uses the replay buffer of a policy trained up to the performance of the medium agent. "Medium-Expert" uses a 50-50 split of medium and expert data.

### 4.3  Findings and Suggestions

**Information-gain-based criteria empirically result in a high switching cost.** The theoretically guided information-gain-based criteria always recover the rewards but suffer from higher switching cost

---

[9]https://github.com/sfujim/BCQ

[10]https://github.com/sfujim/TD3_BC

in practical benchmark tasks with modern RL algorithms. We remark that most theoretical works focus on simplified bandits or tabular MDPs when analysing mathematical properties of information-gain-based methods. These theoretical conclusions are carried out on much simplified scenarios compared with popular deep RL testbeds that we consider here. In addition, these theoretical analyses often ignore the constant factors, which can be crucial for the practical use.

**The difference in the output action distribution may not be a good indicator to measure policy variation.** Many RL works utilize action difference (KL divergence) to measure the difference between policies Schulman et al. (2015); Galashov et al. (2019); Rudner et al. (2021). In our low-switching-cost RL experiments, we empirically observe that the policy-based criterion is particularly sensitive to the threshold parameter. Further analysis (Fig. 3) reveals that the output action changes drastically throughout the training process. Even after the reward converges, we can still observe a significant fluctuation of action difference between the online and the deployed policies. By contrast, feature similarity appears to be more stable. Therefore, we suggest that, when comparing two learning policies in practice, feature-based measurements can be strong candidates for stable policy difference estimates, in addition to popular action-based metrics. We remark that this finding could be applicable to the general RL use cases.

**Feature-based switching criterion empirically performs the best and should be worth more research efforts.** Across all domains, feature-based criterion consistently achieves the lowest switching cost while maintains the similar reward to the case without the low-switching-cost constraint. Feature learning has been widely investigated from the theoretical perspective in other domains, such as supervised learning Allen-Zhu & Li (2020); Du et al. (2021); Tripuraneni et al. (2020), constrastive learning Tian et al. (2021) and RL Agarwal et al. (2020a). However, there is little attention on feature-based methods in the theoretical literature of low-switching-cost RL. We suggest that more research attention should be put on the feature-based direction. We also provide some intuitive justifications for the feature-based criterion using some recent advances in representation learning theory and hope this insight could inspire future research.

To illustrate the insight, we consider the following setting. Suppose we want to learn $f(\cdot)$, a representation function that maps the input to a $k$-dimension vector. We assume we have input-output pairs $(x, y)$ with $y = \langle w, f^*(x) \rangle$ for some underlying representation function $f^*(\cdot)$ and a linear predictor $w \in \mathbb{R}^k$. For ease of presentation, let us assume we know $w$, and our goal is to learn the underlying representation which together with $w$ gives us 0 prediction error. Suppose we have data sets $\mathcal{D}_1$ and $\mathcal{D}_2$. We use $\mathcal{D}_1$ to train an estimator of $f^*$, denoted as $f^1$, and $\mathcal{D}_1 \cup \mathcal{D}_2$ to train another estimator of $f^*$, denoted as $f^{1+2}$. The training method is empirical risk minimization, i.e.,

$$f^1 \leftarrow \min_{f \in \mathcal{F}} \frac{1}{|\mathcal{D}_1|} \sum_{(x,y) \in \mathcal{D}_1} (y - \langle w, f(x) \rangle)^2 \text{ and}$$

$$f^{1+2} \leftarrow \min_{f \in \mathcal{F}} \frac{1}{|\mathcal{D}_1 \cup \mathcal{D}_2|} \sum_{(x,y) \in \mathcal{D}_1 \cup \mathcal{D}_2} (y - \langle w, f(x) \rangle)^2$$

where $\mathcal{F}$ is some pre-specified representation function class.

The following theorem suggests if the similarity score between $f^1$ and $f^{1+2}$ is small, then $f^1$ is also far from the underlying representation $f^*$.

**Theorem 1.** *Suppose $f^1$ and $f^{1+2}$ are trained via aforementioned scheme. There exist dataset $\mathcal{D}_1$, $\mathcal{D}_2$, function class $\mathcal{F}$ and $w$ such that if the similarity score between $f^1$ and $f^{1+2}$ on $\mathcal{D}_{1+2}$ is smaller than $\alpha$, then the prediction error of $f^1$ on $\mathcal{D}_{1+2}$ is at least $1 - \alpha$.*

Theorem 1 suggests that in certain scenarios, if the learned representation has not converged (the similarity score is small), then it cannot be the optimal representation which in turn will hurt the prediction accuracy. Therefore, if the similarity score is small, we should change the deployed policy.

## 5  Conclusion

In this paper, we focus on the low-switching-cost reinforcement learning problem and take the first empirical step towards understanding how to design an effective solution for reducing the switching cost while

maintaining good performance. By systematic empirical studies on practical benchmark environments with modern RL algorithms, we find that there exists a large theory-practice gap in the information-gain-based switching criteria. Meanwhile, the feature-based solution performs the best but has been rarely discussed by the theoretical community. We raise this open question and provide some intuitive justification for the feature-based criterion in this paper. We hope this benchmark project could bring new insights for the community.

**Limitation and Social Impact** We remark that there is still a great research room towards designing a more principled method for low-switching-cost RL. In addition to the feature-based criterion, another important direction is to give provable guarantees for the settings with a large state space or a function approximator beyond existing works on tabular cases (Bai et al., 2019; Zhang et al., 2020b;a). We believe our paper is just the first step on this important problem, which could serve as a foundation towards great future research advances. Medical domains are one potential application for this work, and we believe that it won't result in a worse negative social impact than traditional RL algorithms in this domain. Nevertheless, at the beginning of training, this policy may would not be optimal and should be applied with great caution. A possible direction is to initialize the policy using offline data, and to improve the policy using an online RL algorithm with low switching costs, which we leave as future work.

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

## A Project Statement

All of our code and detailed results can be found at our project website, including installation and running instructions for reproduciblity, and the results of individual games in Atari. The code is hosted at GitHub under the MIT license. We believe that there is not any potential negative societal impact of this project.

All the environments are public accessible RL testbeds. For experiments with MuJoCo engine, we register a free student license. For offline RL experiments, we conduct the experiments based on the codebase and dataset at `https://github.com/aviralkumar2907/CQL` and `https://github.com/google-research/batch_rl`, respectively, which are under Apache-2.0 license. We ensure that such a Atari game dataset does not contain any personally identifiable information or offensive content. All experiments are run on machines with 128 CPU cores, 256G memory, and one GTX 2080 GPU card.

## B Algorithm Details

### B.1 Deep Off-policy Reinforcement Learning:

Deep Q-learning (DQN) (Mnih et al., 2015) is perhaps the most popular off-policy RL algorithm leveraging a deep neural network to approximate $Q(x, a)$. Given the current state $x_h$, the agent selects an action $a_h$ greedily based on parameterized Q-function $Q_\theta(x_h, a)$ and maintain all the transition data in the replay buffer.For each training step, the temporal difference error is minimized over a batch of transitions sampled from this buffer by

$$\mathcal{L}(\theta) = \mathbb{E}\left[(r_{h+1} + \gamma \max_{a'} Q_{\bar{\theta}}(x_{h+1}, a') - Q_\theta(x_h, a_h))^2\right], \tag{5}$$

where $\bar{\theta}$ represents the parameters of the target Q-network, which is periodically updated from $\theta$. Rainbow (Hessel et al., 2018) is perhaps the most famous DQN variant, which combines six algorithmic enhancements and achieves strong and stable performances on most Atari games. In this paper, we adopt a deterministic version[11] of Rainbow DQN as the RL algorithm for the discrete action domains. We also adopt count-based exploration (Tang et al., 2017) as a deterministic exploration bonus.

For continuous action domains, soft actor-critic (SAC) (Haarnoja et al., 2018) is the representative off-policy RL algorithm. SAC uses neural networks parameterized by $\theta$ to approximate both $Q(s, a)$ and the stochastic policy $\pi_\theta(a|s)$. $Q$-network is trained to approximate entropy-regularized expected return by minimizing

$$\mathcal{L}_Q(\theta) = \mathbb{E}[(r_h + \gamma(Q_{\bar{\theta}}(x_{h+1}, a') - \alpha \log \pi(a'|x_{h+1}))$$
$$- Q_\theta(x_h, a_h))^2 | a' \sim \pi(\cdot|x_{h+1})], \tag{6}$$

where $\alpha$ is the entropy coefficient. We omit the parameterization of $\pi$ since $\pi$ is not updated w.r.t $\mathcal{L}_Q$. The policy network $\pi_\theta$ is trained to maximize $\mathcal{L}_\pi(\theta) = \mathbb{E}_{a\sim\pi}[Q(x, a) - \alpha \log \pi_\theta(a|x)]$.

### B.2 Detailed switching criteria

We summarize the switching criteria mentioned in Sec. 3 at Algorithm 2.

## C Experiment Details

### C.1 Environments

**GYMIC** GYMIC is an OpenAI gym environment for simulating sepsis treatment for ICU patients to an infection, where sepsis is caused by the body's response to an infection and could be life-threatening. GYMIC built an environment to simulate the MIMIC sepsis cohort, where MIMIC is an open patient EHR dataset from ICU patients. This environment generates a sparse reward, the reward is set to +15 if the patient recovered and -15 if the patient died. This environment has 46 clinical features and a $5 \times 5$ action space.

---

[11]Standard Rainbow adds random noise to network parameters for exploration, which can be viewed as constantly switching policies over a random network ensemble. This contradicts the low-switching-cost constraint.

---

**Algorithm 2** Switching Criteria ($\mathcal{J}$ in Algorithm 1)

---

▷ Non-adaptive Switching
**input** environment step counter $k$, switching interval $n$
**output** $bool(k \bmod n == 0)$

▷ Policy-based Switching
**input** deployed and online policy $\pi_{\text{dep}}, \pi_{\text{onl}}$, state batch $\mathbb{B}$, threshold $\sigma_p$
Compute the ratio of action difference or KL divergence for $\pi_{\text{dep}}$ and $\pi_{\text{onl}}$ on $\mathbb{B}$ as $\delta$.
**output** $bool(\delta \geq \sigma_p)$

▷ Feature-based switching
**input** Encoder of deployed and online policy $f_{\text{dep}}, f_{\text{onl}}$, state batch $\mathbb{B}$, threshold $\sigma_f$
Compute $sim(\mathbb{B})$ via Eq.(3)
**output** $bool(sim(\mathbb{B}) \leq \sigma_f)$

▷ Visitation-based Switching
**input** the current visited times of state-action pair $n(\phi(x_k), a_k)$
**output** $bool(n(\phi(x_k), a_k) \in \{1, 2, 4, 8...\})$

▷ Information-matrix-based Switching
**input** episode timestep $h$, current covariance matrix $\Lambda_h^e$, old $\widetilde{\Lambda_h^e}$ at previous switch time
Compute the least absolute eigenvalues $v_h^e$ and $\widetilde{v_h^e}$
**output** $bool(v_h^e \geq 2 \times \widetilde{v_h^e})$

---

**Atari 2600**   Atari 2600 games are widely employed to evaluate the performance of DQN based agents. We evaluate the efficiency among different switching criteria on 56 games.

**MuJoCo**   MuJoCo contains continuous control tasks running in a physics simulator, we evaluate different switching criteria on 6 locomotion benchmarks.

For GYMIC and Atari games whose action space is discrete, we adopt Rainbow DQN to train the policy, for MuJoCo tasks with continuous action spaces, we employ SAC since it is more suitable for continuous action space.

### C.2   Hyper-parameters of deterministic Rainbow

Table 2 lists the basic hyper-parameters of Rainbow. All of our experiments share these hyper-parameters except that the experiments on GYMIC adopt $H_{\text{target}} = 1K$. Most of these hyper-parameters are the same as those in the original Rainbow algorithm. For count-based exploration, the bonus $\beta$ is set to 0.01.

Table 3 lists the extra hyper-parameters for experiments on GYMIC. Since there are 46 clinical features in this environment, we stack 4 consecutive states to compose a 184-dimensional vector as the input for the state encoder. The state encoder is a 2-layer MLP with hidden size 128.

Table 4 shows the additional hyper-parameters for experiments on Atari games. The observations are grey-scaled and resized to tensors with size $84 \times 84$, and 4 consecutive frames are concatenated as a single state. Each action selected by the agent is repeated for 4 times. The state encoder is composed of 3 convolutional layers with 32, 64 and 64 channels, which use 8x8, 4x4, 3x3 filters and strides of 4, 2, 1, respectively.

### C.3   Hyper-parameters of SAC

We list the hyper-parameters of SAC in Table 5. We adopt Adam as the optimizer, the learning rate is set to 0.001 for MuJoCo control tasks except for Swimmer which is 0.0003, and the temperature parameter $\alpha$ is set to 0.2 for Mujoco control tasks except for Humanoid which is 0.05. Other hyper-parameters are the

| Parameter | Value |
|---|---|
| $H_{\text{start}}$ | 20K |
| learning rate | 0.0000625 |
| $H_{\text{target}}(\text{Atari})$ | 8K |
| $H_{\text{target}}(\text{GYMIC})$ | 1K |
| Adam $\epsilon$ | $1.5 \times 10^{-4}$ |
| Prioritization type | proportional |
| Prioritization exponent $\omega$ | 0.5 |
| Prioritization importance sampling | $0.4 \rightarrow 1.0$ |
| Multi-step returns n | 3 |
| Distributional atoms $N_{\text{atoms}}$ | 51 |
| Distributional $V_{\text{min}}, V_{\text{max}}$ | [-10, 10] |
| Discount factor $\gamma$ | 0.99 |
| Memory capacity N | 1M |
| Replay period | 4 |
| Minibatch size | 32 |
| Reward clipping | [-1, 1] |
| Count-base bonus | 0.01 |
| Activation function $\beta$ | ReLU |

Table 2: The basic hyper-parameters of Rainbow. we used the Adam optimizer with learning rate $\alpha = 0.0000625$ and $\epsilon = 1.5 \times 10^{-4}$ . Before training the online policy, we let the initialized random policy run 20K steps to collect some transitions. The capacity of the replay buffer is 1M. During the training process, we sample 32 transitions from the replay buffer and update the online policy every four steps. The reward is clipped into [-1, 1] and ReLU is adopted as the activation function. For replay prioritization we use the recommended proportional variant, with importance sampling from 0.4 to 1, the prioritization $\omega$ is set to 0.5. In addition, we employ $N_{\text{atoms}} = 51, V_{\text{min}} = -10, V_{\text{max}} = 10$ for distributional RL and $n = 3$ for multi-step returns. The count-based bonus is set to 0.01.

| Parameter | Value |
|---|---|
| State Stacked | 4 |
| Number of layers for MLP | 2 |
| Hidden size | 128 |

Table 3: Extra hype-parameters for the experiments in GYMIC, we stack 4 consecutive states and adopt a 2-layer MLP with hidden size 128 to extract the feature of states.

same for these tasks. The update frequency "50/50" means we perform 50 iterations to update the online policy per 50 environment steps.

## C.4 Hyper-parameters of different criteria

For the switching threshold in policy-based and feature-based criteria (i.e., the mismatch ratio $\sigma_p$ in policy-based criterion and parameter $\sigma_f$ in feature-based criterion), we perform a rough grid search and choose the highest possible threshold that still produces a comparable final policy reward. For GYMIC, we tried $\sigma_p \in \{0.25, 0.5\}$ and $\sigma_f \in \{0.97, 0.98, 0.99\}$ and finally adopted $\sigma_p = 0.5$ and $\sigma_f = 0.97$. For Atari games, we tried $\sigma_p \in \{0.25, 0.5\}$ and $\sigma_f \in \{0.98, 0.99\}$. For MuJoCo, we tried $\sigma_p \in \{0.5, 1.0, 1.5\}^{12}$ and $\sigma_f \in \{0.7, 0.8, 0.9\}$

---

[12]$\{0.5, 1.0, 1.5\}$ is the thresholds of KL.

| Parameter | Value |
|---|---|
| Gray scaling | True |
| Observation | (84, 84) |
| Frame Stacked | 4 |
| Action repetitions | 4 |
| Max frames per episode | 108k |
| Encoder channels | 32, 64, 64 |
| Encoder filter size | $8 \times 8, 4 \times 4, 3 \times 3$ |
| Encoder stride | 4, 2, 1 |

Table 4: Additional hyper-parameters for experiments in Atari games. Observations are grey-scaled and rescaled to $84 \times 84$. 4 consecutive frames are staked as the state and each action is acted for four times. We limit the max number of frames in an episode to 108K. The state encoder consists of 3 convolutional layers.

| Parameter | Value |
|---|---|
| Warm-up samples | 10K |
| Learning rate | 0.001 |
| Optimizer | Adam |
| Temperature parameter $\alpha$ | 0.2 |
| Discount factor $\gamma$ | 0.99 |
| Memory capacity N | 1M |
| Number of hidden layers | 2 |
| Number of hidden nodes per layer | 256 |
| Target smoothing coefficient | 0.005 |
| Update frequency | 50/50 |
| Target update interval | 1 |
| Minibatch size | 128 |
| Activation function | ReLU |

Table 5: Hyper-parameters of SAC algorithms on MuJoCo control tasks.

## D  Discussion on RSI

### D.1  RSI without log function

We choose log function because we observe that switching costs of different criteria vary in orders of magnitudes. We also tried a variant of RSI that removes the log function, which is

$$\mathbb{I}\left[R_{\mathcal{J}} > \left(1 - \text{sign}(\hat{R})\sigma_{\text{RSI}}\right)\hat{R}\right]\left(\max\left(\frac{\hat{C}}{C_{\mathcal{J}}}, 1\right)\right), \tag{7}$$

The results are shown in the following Table 6 ($\sigma_{RSI} = 0.2$ ). We think these numbers have exaggerated the differences among different criteria and we finally adopt the log function.

| | Feature | Policy | Info | Visitation | Fix_1000 |
|---|---|---|---|---|---|
| GYMIC | **10943** | 51 | 4408 | 1 | 1000 |
| Atari | **1028** | 184 | 93 | 57 | 643 |
| MuJoCo | **10826** | 136 | 128 | 10 | 833 |

Table 6: The results of RSI variant that removes the log function, the results exaggerate the differences among different criteria.

## D.2 Evaluate RSI with different $\sigma_{\mathbf{RSI}}$

we evaluate RSI using different $\sigma_{RSI}$, and list the results in Table 7. Since the rewards obtained by different criteria on GYMIC are almost the same, RSI remains mostly unchanged when using different $\sigma_{RSI}$. The only exception is when $\sigma_{RSI} = 0$. In this case, RSI of "Feature" and "Info" become 0. This is because a positive RSI requires strictly better or equal reward performance than "None" criterion when $\sigma_{RSI} = 0$. For Atari and MuJoCo, the overall trend is that a larger $\sigma_{RSI}$ results in a larger RSI, since a larger $\sigma_{RSI}$ tolerates a wider performance range. And we can observe that for most $\sigma_{RSI}$, the conclusions remain unchanged.

| GYMIC | | | | | | | | | | |
|---|---|---|---|---|---|---|---|---|---|---|
| $\sigma_{\mathrm{RSI}}$ | 0.0 | 0.1 | 0.2 | 0.3 | 0.4 | 0.5 | 0.6 | 0.7 | 0.8 | 0.9 |
| Feature | 0.0 | **9.30** | **9.30** | **9.30** | **9.30** | **9.30** | **9.30** | **9.30** | **9.30** | **9.30** |
| Policy | 3.93 | 3.93 | 3.93 | 3.93 | 3.93 | 3.93 | 3.93 | 3.93 | 3.93 | 3.93 |
| Info | 0.0 | 8.39 | 8.39 | 8.39 | 8.39 | 8.39 | 8.39 | 8.39 | 8.39 | 8.39 |
| Visitation | 0.0 | 0.0 | 0.0 | 0.0 | 0.0 | 0.0 | 0.0 | 0.0 | 0.0 | 0.0 |
| FIX_1000 | **6.91** | 6.91 | 6.91 | 6.91 | 6.91 | 6.91 | 6.91 | 6.91 | 6.91 | 6.91 |

| Atari | | | | | | | | | | |
|---|---|---|---|---|---|---|---|---|---|---|
| RSI | 0.0 | 0.1 | 0.2 | 0.3 | 0.4 | 0.5 | 0.6 | 0.7 | 0.8 | 0.9 |
| Feature | 1.68 | **3.06** | **4.00** | 3.90 | 4.50 | 4.70 | **5.02** | **5.13** | **5.42** | **5.54** |
| Policy | 1.60 | 2.74 | 3.22 | 3.66 | 4.15 | 4.50 | 4.81 | 4.81 | 4.81 | 4.88 |
| Info | 0.81 | 1.65 | 2.33 | 3.14 | 3.62 | 3.84 | 4.01 | 4.17 | 4.36 | 4.45 |
| Visitation | 0.84 | 1.14 | 1.83 | 2.02 | 2.11 | 2.11 | 2.17 | 2.17 | 2.17 | 2.25 |
| FIX_1000 | **1.87** | 2.96 | 3.25 | **4.24** | **4.63** | **4.83** | 4.93 | 4.93 | 5.03 | 5.13 |

| MuJoCo | | | | | | | | | | |
|---|---|---|---|---|---|---|---|---|---|---|
| RSI | 0.0 | 0.1 | 0.2 | 0.3 | 0.4 | 0.5 | 0.6 | 0.7 | 0.8 | 0.9 |
| Feature | 2.55 | **7.66** | **8.26** | **8.26** | **8.26** | **8.26** | **8.26** | **8.26** | **8.26** | **8.26** |
| Policy | 2.64 | 4.25 | 5.26 | 6.01 | 6.01 | 6.01 | 6.01 | 6.01 | 6.01 | 6.01 |
| Info | 2.96 | 4.53 | 4.69 | 4.83 | 4.83 | 4.83 | 4.83 | 4.83 | 4.83 | 4.83 |
| Visitation | 1.23 | 1.86 | 1.90 | 1.94 | 1.94 | 1.94 | 1.94 | 1.94 | 1.94 | 1.94 |
| FIX_1000 | **4.60** | 6.22 | 6.68 | 6.91 | 6.91 | 6.91 | 6.91 | 6.91 | 6.91 | 6.91 |

Table 7: The RSI of different $\sigma_{\mathrm{RSI}}$

# E Proof of Theorem 1

*Proof.* We let $w = (1, 1, \ldots, 1) \in \mathbb{R}^k$ be a $k$-dimensional all one vector. We let

$$\mathcal{F} = \{f : f(x) = (2\sigma(\langle v_1, x \rangle) - 1, 2\sigma(\langle v_2, x \rangle) - 1, \ldots, $$
$$2\sigma(\langle v_k, x \rangle) - 1)\} \subset \{\mathbb{R}^k \to \mathbb{R}^k\}$$

with $\sigma(\cdot)$ being the ReLU activation function[13] and $v_i \in \{e_i, -e_i\}$ where $e_i \in \mathbb{R}^k$ denotes the vector that only the $i$-th coordinate is 1 and others are 0. We assume $k$ is an even number and $\alpha k$ is an integer for simplicity. We let the underlying $f^*$ be the vector correspond to $(e_1, e_2, \ldots, e_k)$. We let $\mathcal{D}_1 = \{(e_1, 1), (e_2, 1), \ldots, (e_{(1-\alpha)k}, 1)\}$ and $\mathcal{D}_2 = \{(e_{(1-\alpha)k+1}, 1), \ldots, (e_k, 1)\}$. Because we use the ERM training scheme, it is clear that the training on $\mathcal{D}_1 \cup \mathcal{D}_2$ will recover $f^*$, i.e., $f^{1+2} = f^*$ because if it is not $f^*$ is better solution ($f^*$ has 0 error) for the empirical risk. Now if the similarity score between $f^1$ and $f^{1+2}$ is smaller than $\alpha$, it means for $f^1$, its corresponding $\{v_{(1-\alpha)k+1}, \ldots, v_k\}$ are not correct. In this case, $f^1$'s prediction error is at least $1 - \alpha$ on $\mathcal{D}_1 \cup \mathcal{D}_2$, because it will predict 0 on all inputs of $\mathcal{D}_2$.

□

---

[13] We define $\sigma(0) = 0.5$

