# OpenReview forum: "Beyond Information Gain: An Empirical Benchmark for  Low-Switching-Cost Reinforcement Learning"
_TMLR — Accepted by TMLR_

### Review · Reviewer_wNAL · 2022-10-18

**Summary Of Contributions:**

The paper considers the Reinforcement Learning setting in which the behavior policy should change (be updated to the currently-learned policy) as little often as possible. A motivation is given for when the behavior policy is implemented in an FPGA, in which case deploying a new policy can take a few hours. In this low-switching-cost RL framework, the authors review several approaches at choosing when to deploy a new policy, identify that information-gain-based approaches outperform the other ones, and propose a scalable feature-based way to approximate information gain, based on a matrix built on features, and analysis of its eigen values.

**Audience:**

Yes

**Broader Impact Concerns:**

No broader impact concern

**Claims And Evidence:**

Yes

**Requested Changes:**

- A strong comparison against several recent batch RL methods (details about how the dataset are generated can be put in the appendix). This change is mainly to strengthen the paper, I think it is already acceptable as is.

**Strengths And Weaknesses:**

Strong points:

- Low-switching-cost RL is relatively well-presented and motivated in the introduction, even though I suggest later a dedicated section on motivating low-switching-cost RL (just for even better readability)
- The paper is well-written and clear. The algorithms being compared are quite easy to understand from the paper.
- The proposed method seems original, even if the paper mainly claims to be empirically comparing existing methods.
- The experiments seem well-designed and consider real-world tasks (GYMIC). The fact that there are also Atari games and Mujoco tasks indicates a very general approach, and broad comparison, on many kinds of environments (with image observations, and continuous actions)
- The comparison against CQL (batch RL) is welcome

Weak points:

- The definition of low-switching-cost RL is much clearer when read in combination with the examples given in the introduction. I think that listing some examples in Section 3.1 (very briefly) would help the reader understand what we try to optimize in this paper.
- I did not find in the paper an explanation of how the dataset for CQL was generated, whether this was tune, and what happens if some other batch RL algorithms are used (Batch-Constrained Q for instance). Some recent paper on batch RL [1] seems interesting on its own, and lists much related work on batch RL.

[1]: Fujimoto, S., & Gu, S. S. (2021). A minimalist approach to offline reinforcement learning. Advances in neural information processing systems, 34, 20132-20145.

---

> ### Author Response · Authors · 2022-10-25
> **We have updated our paper. All the changes are highlighted in red.**
>
> Thanks for your careful review and helpful feedback.
>
> We modify section 3.1 and list some examples to clarify what we are trying to optimize.
>
> For CQL experiments on the Atari games, we re-run the author-provided code in https://github.com/aviralkumar2907/CQL and use the same dataset with the author. The dataset comes from https://github.com/google-research/batch_rl, which was collected during the DQN training process.
>
> We further compare the low-switching-cost setting against several recent offline RL methods (CQL, BCQ, TD3+BC) on MuJoCo tasks. We use the author-provided implementations for the offline algorithms and evaluate these methods on the D4RL benchmark. We also apply
> the feature-based switching criteria to TD3. For TD3 (online) and TD3+Feature, we remove the behavior cloning regularization term and the feature-normalizing process in the codebase of TD3+BC.
>
> The results in Fig. 6 indicate that the performances of offline methods are highly dependent on the dataset's quality. No offline algorithm obtains comparable performance with online training on the "random" or "medium-replay" dataset. However, the low-switching-cost setting does not need a dataset. It can recover online training performances and reduce switching costs by orders of magnitude. The focus of offline RL is parallel to the low-switching-cost setting.
>
> We have updated our paper. Please refer to Fig.6 for the new results.

---

### Review · Reviewer_yPPd · 2022-10-27

**Summary Of Contributions:**

This paper considers low-switching-cost RL setting, i.e., achieving the highest reward while reducing the number of policy switches during training. It has been a recent trend in theoretical RL research to develop provably efficient RL algorithms with low switching cost. The core idea in these theoretical works is to measure the information gain and switch the policy when the information gain is doubled. This paper conducts the first empirical evaluation of different policy switching criteria on popular RL testbeds, including a medical treatment environment, the Atari games, and robotic control tasks. Surprisingly, although information-gain-based methods do recover the optimal rewards, they often lead to a substantially higher switching cost. By contrast, this paper find that a feature-based criterion, which has been largely ignored in the theoretical research, consistently produces the best performances over all the domains.

**Audience:**

Yes

**Broader Impact Concerns:**

No.

**Claims And Evidence:**

Yes

**Requested Changes:**

One weakness I already mentioned in above. Another aspect is a related paper [1] is not discussed. I urge the authors to include this related paper in the revision.

[1] Sample-Efficient Reinforcement Learning with loglog(T) Switching Cost, ICML 2022.

**Strengths And Weaknesses:**

Strength: This paper conducts the first empirical study for low-switching-cost RL on environments that require modern RL algorithms, i.e., Rainbow (Hessel et al., 2018) and SAC (Haarnoja et al., 2018), including a medical environment, 56 Atari games1 and 6 MuJoCo control tasks. This paper finds that a feature-based criterion produces the best overall quantitative performance, which largely outperforms the theoretically guided ones based on information gain.

Weakness: The paper seems has already been edited with red font. Overall, the result relatively good to me. The only thing I am concerning is that the Non-adaptive switching, policy-based switching, and feature-based switching seems are heuristic approaches and it would nice if some theoretical guidance can be conducted to validate those criteria, for example, under some weak assumptions, convergence of low-switching can be guaranteed.

---

> ### Comment · Reviewer_yPPd · 2022-10-31
> **Re**
>
> Thank you for the update. It looks nice to me.

---

### Review · Reviewer_5Vfo · 2022-11-10

**Summary Of Contributions:**

This paper is an empirical study on the topic of low-switching cost RL — which is an RL paradigm that seeks to maximize the return of the agent as well as the number of times the deployed policy which interacts with the environment switches during training. The authors tested a wide range of switching criteria on a set of popular DRL benchmarks provided evidence on the empirical performance of these criteria both in terms of minimizing the switching cost and maximizing the reward.

**Audience:**

Yes

**Broader Impact Concerns:**

As suggested by the authors, one potential application of this work is within medical domains where low switching cost is an essential criterion for policy deployment. I suggest the authors further discuss the broader ethical and societal impacts of this application.

**Claims And Evidence:**

No

**Requested Changes:**

As I mentioned previously, due to the empirical nature of this work, 3 seeds is too small to obtain any conclusive results, I suggest doing the experiments with a few more seeds (5 should be a bare minimum). My other suggestions are listed above

**Strengths And Weaknesses:**

This is an interesting work which I believe may provide some useful insights for the RL community. Low-switching-cost RL is an important setting in many potential applications and gaining a better understanding of the empirical performance of common algorithms/paradigms can be a major step forward in landing some real-world impact for the field. My comments/questions below:
- In the second to last paragraph in Section 3.2 (visitation-based switching), I suggest adding citation(s) related random projection and hashing.
- 3 seeds is too small, I strongly recommend 5 at the very least (ideally 10). I think this is a critical to this paper since the work itself is empirical and it is important that we are given strong empirical evidence to support the conclusions. This is particularly problematic for the GYMIC environment which showed high variance across seeds.
- Any ideas why non-adaptive methods worked so well on Atari?
- Several questions related to Section 4.3
  - Could you give some intuition or any conjectures you may have on the theoretical-practical discrepancies of information-gain based methods? One thing you mentioned is that the constant factor often gets ignored, do you have any sense what may influence the constant factor?
  - In this work, you take the last layer of the Q network as the features and compute a similarity score. Since the policies themselves are induced (either directly or indirectly) by the Q network, on a high level, feature-based methods and policy-based methods are of a similar flavor. Could you comment on your intuition on why the former is more stable compared to the latter? I think overall some discussion on the connection between the three adaptive approaches (information gain, feature-based, policy-based) would be a nice touch
- Could you give some brief comments on the infinite horizon setting? For example, are there any particular challenges in extending this work to the infinite horizon case?

---

> ### Author Response · Authors · 2022-11-24
> **Thanks for your careful review and helpful feedback. We have conducted experiments over more seeds and updated our paper.**
>
> We have added citations related to random projection and hashing to Section 3.2.
>
> For the Atari and Mujoco environments, we conducted experiments over 2 more seeds and the conclusions remain unchanged. For the GYMIC experiment, we find that the performance variance over different episodes is high. Therefore we increase the number of test episodes from 100 to 1,000 when evaluating the performance of checkpoints, and all experiments are repeated over 10 seeds in this environment. Now the performance variance in GYMIC seems non-problematic, and the conclusion remains unchanged. We have updated the results in Fig.1, Fig.2, Fig.4, Fig.5, Tab.1, Tab.6, and Tab.7.
>
> For the non-adaptive method, we want to emphasize that the feature-based method outperforms the non-adaptive method significantly in most environments, but the switching cost curve in the log scale obscures the difference. Additionally, we tried 100, 1,000, and 10,000 as switching intervals. According to our results, FIX-10000 cannot recover the performance of FIX-100, and FIX-100 causes the switching costs to be higher than FIX-1000. You can refer to https://sites.google.com/view/low-switching-cost-rl/mujuco and https://sites.google.com/view/low-switching-cost-rl/atari for the ablation study results. It is also an interesting finding for us that the non-adaptive method performs so well, but this is highly related to the environment.
>
> Most theoretical works focus on simplified bandits or tabular MDPs when analyzing the mathematical properties of information-gain-based methods, and achieve the switching cost associated with the number of states and actions as well as the episodic horizon (e.g., O(HSAlog(T)), S is the number of states, A denotes the number of actions, and H is the horizon of an episode). Deep RL testbeds we consider have large S (the states are continuous and action spaces are extremely large). We think this is one reason for theoretical-practical discrepancies.
>
> We realize that the sentence “*we take the final hidden layer of the Q-network as the feature representation*” is misleading. We have modified this sentence. The main differences of the feature-based and policy-based methods are listed as follows:
>
> - **The policy-based and feature-based methods are based on different layers of the network.** For the Atari game, we use the convolution layers to process the image inputs and adopt the output of the final convolution layer as the feature representation. The feature representation is then fed into an MLP and the action is chosen using “argmax”. For other environments, the situation is similar. **There is an MLP with a single hidden layer between the feature representation we adopted and the final output of the Q-network.**
> - The feature-based criterion is based on a high-dimensional representation, while the policy-based criterion is based on several actions. We hypothesize that the actions would change more because of their direct association with the training objective. And any change in high-dimensional features may have a significant impact on actions.
> - The results in Fig.3 validate that the feature-based criterion is more stable.
>
> The feature-based, policy-based, visited-based and non-adaptive methods can be extended to infinite horizons without particular challenges. However, the information-matrix-based method cannot be extended to infinite horizons, since this method compares the information-matrix of the same steps among **different** episodes. And theoretically, the switching cost of the Information-matrix-based method is related to the horizon (e.g., O(HSAlog(T)), which is not suitable for infinite horizon setting.
>
> **Limitation and Social Impact**
> Medical domains are one potential application for this work, and we believe that it won't result in a worse negative social impact than traditional RL algorithms in this domain. Nevertheless, at the beginning of training, this policy may would not be optimal and should be applied with great caution. A possible direction is to initialize the policy using offline data, and to improve the policy using an online RL algorithm with low switching costs, which we leave as future work.  We have discussed this in the conclusion section.

---

> > ### Comment · Reviewer_5Vfo · 2022-12-13
> > **Response to review**
> >
> > Thank you so much for your detailed response, I went through your updated paper as well as the other reviews, overall I am satisfied with the changes you have made and your response to mine and other reviewers' concerns.

---

### Decision · Action_Editors · 2022-12-31

**Recommendation:** Accept as is

**Comment:**

The paper studies the reinforcement learning setting in which the policy should change as little as possible during training.
The authors empirically evaluate several criteria for choosing when switching the policy on a set of deep reinforcement learning benchmarks.
The reviewers agree that this paper addresses a relevant topic and that the empirical findings in the paper are significant.
The authors' responses have adequately addressed the issues raised in the reviews.
The new version of the paper has been considered satisfactory by the reviewers, who agreed to accept this paper.

**Audience:**

The paper deals with an important but overlooked setting in the Reinforcement Learning field, which is of interest to a significant part of the TMLR's audience.

**Claims And Evidence:**

The claims in the paper are supported by an extensive empirical evaluation that provides clear evidence.